# Effect of Graphene on the Microstructure and Mechanical Properties of WC-Based Cemented Carbide

**Wanzhen Qi, Zhiwei Zhao \*, Yanju Qian, Shijie Zhang, Hongjuan Zheng, Xiaomiao Zhao, Xinpo Lu and Shun Wang**

College of Materials Science and Engineering, Henan University of Technology, Zhengzhou 450001, China; 15639427067@163.com (W.Q.); qianyanju001@163.com (Y.Q.); shijie_zhang@haunt.edu.cn (S.Z.); hongjuan_zheng@haunt.edu.cn (H.Z.); xiaomiao_zhao@haut.edu.cn (X.Z.); xinpo_lu@haut.edu.cn (X.L.); shun_wang@haut.edu.cn (S.W.)
\* Correspondence: zhiwei_zhao@haut.edu.cn; Tel.: +86-371-67758730; Fax: +86-371-67758729

**Abstract:** WC-based cemented carbides were prepared by spark plasma sintering (SPS) of WC-Co-Cr$_3$C$_2$-VC alloy powder by adding different contents of graphene. The phase composition, microstructure, mechanical properties, and magnetic properties of cemented carbide were investigated by means of XRD, SEM, Vickers hardness and fracture toughness tests, and magnetic properties tests. The results showed that the mechanical properties of the specimens show a trend of first increasing and then decreasing with the increase in graphene content. After adding 0.6 wt.% graphene, graphene is uniformly distributed on the substrate in the form of flakes, WC grain size decreases, the hardness of the specimen increases to 2009 HV, the relative density increases to 94.3%, the fracture toughness is 11.72 MPa·m$^{1/2}$, and the coercivity of the sample is 437.55 Oe. Therefore, cemented carbide with a graphene content of 0.6 wt.% has excellent comprehensive performance (Vickers hardness and fracture toughness).

**Keywords:** cemented carbides; spark plasma sintering; graphene; mechanical properties

## 1. Introduction

Cemented carbide is a composite material made by powder metallurgy with refractory metal WC powder as the base and the introduction of transition group metal Co as the sintering bonding phase [1–3]. Among them, the WC phase is the hard phase, which contributes to the hardness and wear resistance of the substrate and is the first and most common refractory carbide used in cemented carbide [4]. Cemented carbide, metal ceramics, and high-speed steel are commonly used metal tool materials. Among these materials, cemented carbide is widely used because of its high hardness, wear resistance, and certain toughness, especially in wear-resistant parts, tools [5], mining processing, and so on. Despite the excellent mechanical properties of cemented carbide, traditional cemented carbide can no longer meet the realistic requirements of high hardness and high toughness because the hardness and fracture toughness of cemented carbide are always contradictory to each other [6–8]. Studies have shown that the finer the WC grain size in cemented carbide, the greater its hardness and toughness [9–11]. Therefore, how to obtain WC-based cemented carbide with high hardness and high toughness has become an essential research direction.

In recent years, several researchers have improved the mechanical properties of cemented carbide by adding rare earth elements, second-phase oxides, and carbon nanomaterials, such as Re [12], nano-CeO$_2$ [13], nano-Al$_2$O$_3$ [14], Cr$_3$C$_2$ [15], graphene [16], and so on. Among them, graphene, as a typical two-dimensional nanomaterial, has attracted great attention due to its excellent mechanical properties, high-temperature stability, thermal conductivity, and other properties, which improve the mechanical properties of ceramic materials [17–19]. Moreover, from the perspective of dynamics, the epitaxial growth of heterogeneous structures of 2D graphene materials has the potential to make it a more excellent material with comprehensive properties such as mechanical properties and thermal

stability [16,20]. Research [21] shows that adding a certain amount of graphene platelet (GPL) to cemented carbide can refine the WC grains, thereby improving the hardness, toughness, and wear resistance of the composite. Other researchers [4,6] have also come to a similar conclusion that adding an appropriate amount of graphene to cemented carbide can improve its mechanical properties.

The main research of this experiment is to develop cemented carbide with high hardness and toughness by introducing graphene as a reinforcing phase, using $Cr_3C_2$-VC as a composite grain growth inhibitor (GGIs). Graphene-reinforced WC-Co-$Cr_3C_2$-VC cemented carbide was prepared by the spark plasma sintering (SPS) process. Graphene can inhibit the grain growth of WC to some extent and improve the toughness of the cemented carbide. As a rapid sintering method, spark plasma sintering (SPS) has the advantages of fast sintering speed, short sintering time, low sintering temperature, uniform grain size, and high densification degree compared with the traditional sintering method [22–24]. In addition, composite grain inhibitors can effectively inhibit the grain growth of WC, thereby refining the grain size of cemented carbide. Therefore, the raw materials of cemented carbide can be densified at lower temperatures (1200~1350 °C) and in a shorter time (5~10 min). In this experiment, WC-Co-$Cr_3C_2$-VC and graphene were used as raw materials to prepare WC-based cemented carbide by the SPS sintering process. The effects of graphene content on the phase composition, mechanical properties, magnetic properties, and microstructure of the cemented carbide were studied.

## 2. Materials and Methods

### 2.1. Material Preparation

Using nano-WC (purity > 99.9%, average particle size < 200 nm, Shanghai Shuitian Material Technology Co., Ltd., Shanghai, China), nano-VC (purity > 99.9%, average particle size < 200 nm, Shanghai Shuitian Material Technology Co., Ltd., Shanghai, China), nano-$Cr_3C_2$ (purity > 99.9%, average particle size < 100 nm, Shanghai Shuitian Material Technology Co., Ltd., Shanghai, China), nano-Co (purity > 99.9%, average particle size < 50 nm, Shanghai Shuitian Material Technology Co., Ltd., Shanghai, China), and graphene nanosheets (GNSs) (purity > 95%, chip diameter 1–3 mm, thickness 1–5 nm, Nanjing Xianfeng, Nanjing, China) as raw materials, according to a specific mass percentage (WC: VC: $Cr_3C_2$: Co = 93.5 wt.%: 0.25 wt.%: 0.25 wt.%: 6.0 wt.%), graphene with different mass percentages (0.2 wt.%, 0.4 wt.%, 0.6 wt.%, 0.8 wt.%) was weighed and added, mixing via the QM-3SP2 planetary ball mill (Nanjing Laibu Technology Industry Co., Ltd., Nanjing, China), ball to material ratio 4:1, ball milling medium anhydrous ethanol, and ball milling at a speed of 180 rmp for 24 h. The mixture was then placed in a drying oven at 80 °C for 12 h, and then the composite was sintered in an SPS-30 spark plasma sintering furnace (Shanghai Chenxin Electric Furnace Co., Ltd., Shanghai, China) at a sintering temperature of 1350 °C, a holding time of 8 min, and a sintering pressure of 50 MPa.

### 2.2. Characterization

An X-ray diffraction analyzer type D8AA25 (Bruker AXS, Karlsruhe, Germany), introduced from Germany, was used for the physical phase analysis of WC matrix composites with Cu-K$\alpha$ in the 2 theta angle range of $20° \leq 2\theta \leq 90°$. An INSPECT F50 scanning electron microscope (SEM) (FEI, Hillsboro, OR, USA) and energy dispersive spectrometer (EDS) (FEI, Hillsboro, OR, USA) introduced by the American FEI Company were used to characterize the micromorphology and surface elements of the samples. The surface morphology and structure of the product were further observed by Oxford Instruments MFP-3D infinite atomic force microscope (Oxford Instruments, Les Ulis, France), FEI-TALOS-F200X transmission electron microscope (FEI, Hillsboro, OR, USA), and super-X energy spectrometer (FEI, Hillsboro, OR, USA). The chemical composition and binding state of the samples were probed by an XSAM800 X-ray photoelectron spectrometer from Kratos, UK. The hysteresis loops and magnetic properties of specimens were acquired using a vibrating sample magnetometer (JDAW-2000D, Jilin University, Jilin, China).

The density of the specimen was measured utilizing the Archimedes principle using the ED-300A digital solid density meter. Hardness measurements were developed using a Vickers hardness tester (FM-ARS900, Future-Tech, JPN) with a loading rate of 50 μm/s and a load of 1 kgf, and the results were averaged over 5 random points in each sample. The fracture toughness of the alloy was calculated from the measured crack length after the indentation test using the following Formula (1) [25]:

$$K_{IC} = 0.0028(\text{Hv})^{1/2}(\text{P/L})^{1/2} \tag{1}$$

where $K_{IC}$ is the fracture toughness (MPa·m$^{1/2}$), Hv is the Vickers hardness (kgf/mm$^2$ or N/mm$^2$), L is the total length of crack extension (mm), and P is the applied load (kgf or N).

## 3. Results and Discussion

### 3.1. Phase Composition and Binding State

Figure 1 shows the XRD patterns of the WC-Co-Cr$_3$C$_2$-VC cemented carbide with different graphene contents added at 1350 °C, 8 min, and 50 MPa. As shown in Figure 1, the WC-based cemented carbide prepared by SPS is mainly composed of two phases, a WC phase and a Co phase, with no diffraction peaks representing Cr$_3$C$_2$ and VC due to the experimentally added grain inhibitor content being below the detection range of XRD. As can be seen from the inset of Figure 1, with the increase in graphene content, the peak value of WC shifts first to the right and then to the left slightly. When the graphene content is increased from 0.0 wt.% to 0.4 wt.%, the peak value of WC shifts to the right slightly, according to Bragg's equation, and the interplanar crystal spacing of WC grains decreases. When the graphene content is 0.6 wt.%, the peak value begins to shift to the left, indicating that after the addition of graphene, the crystal face spacing of the WC grains increases. This is because the solid solution strengthening effect occurs after graphene is added. At a graphene content of 0.8 wt.%, the intensity of the diffraction peak decreases, and a diffraction peak of phase C appears, which should be the graphene phase, probably due to the agglomeration of excess graphene, increasing the intensity of the diffraction peak. The accumulated graphene will reduce the refining effect on the WC grains and then affect the mechanical properties of cemented carbide. In agreement with Sun et al.'s studies [26–28], no harmful compounds such as η-phase and free carbon were detected in all samples, indicating that the graphene and carbon content in the matrix is appropriate. Other authors [29] also reported that the SPS technique has a positive effect on the suppression of the formation of unfavorable phases in the WC-Co system.

To investigate the chemical composition and bonding state of WC-based cemented carbide with graphene content of 0.6 wt.% prepared under sintering conditions of 50 MPa, 1350 °C, and 8 min, an XPS examination was carried out; this case is depicted in Figure 2. As shown in Figure 2a, the WC-based cemented carbide is mainly composed of six elements: C, W, Co, O, Cr, and V. Figure 2b shows the XPS spectrum of the C1s energy region of the sample, which consists of four main peaks. The binding energy of peak A is 280.05 eV, corresponding to WC. The binding energy of peak B is 284.75 eV, which should be the free carbon in the form of graphite polluting the carbide surface. Peak C (286.05 eV) and peak D (287.85 eV) may correspond to carbides of W, V, or Cr. As shown in Figure 2c, peak A (577.6 eV) and peak B (587.2 eV) correspond to oxides and carbides of Cr, respectively. Peak A (31.5 eV) in Figure 2d corresponds to WC, while peak B (33.65 eV) and peak C (35.3 eV) correspond to WO$_3$. The energy of peak D is 37.45 eV, which corresponds to the other compounds of W (W(OPh)$_6$, etc.).

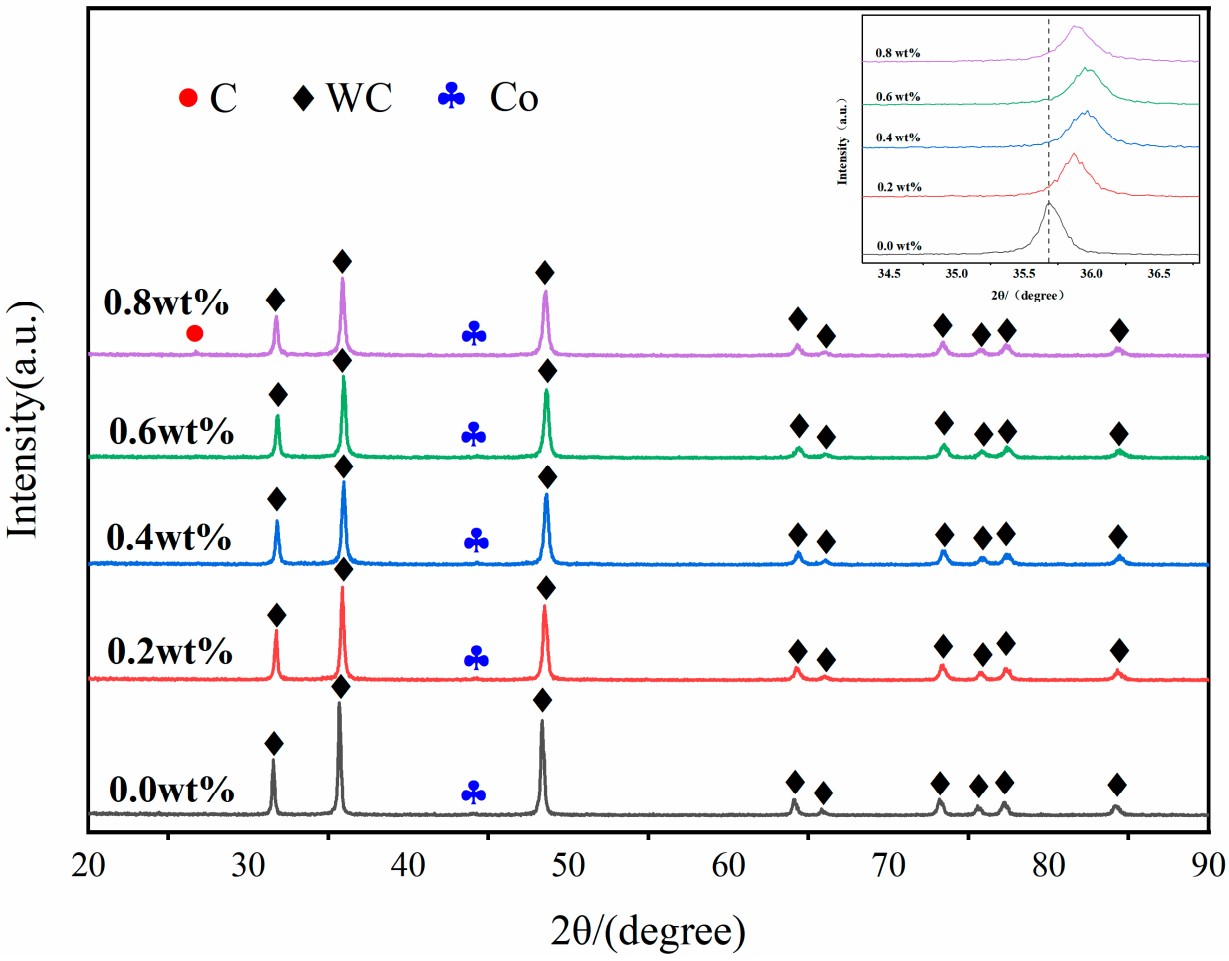

**Figure 1.** XRD patterns of cemented carbide with different graphene contents at 1350 °C, 8 min, and 50 MPa.

As can be seen from Figure 2e, peak A (530.9 eV) corresponds to $WO_3$ or CoO, which is a phenomenon caused by oxidation of the sample surface. Similar phenomena have been reported in other studies [30]. There is only one more prominent peak whose bending energy is 522.1 eV in Figure 2f, which corresponds to the VC. As shown in Figure 2g, the bending energy of peak A is 781.8 eV, which corresponds to CoO, and peak B (786.8 eV), peak C (797.6 eV), and peak D (803.2 eV) all correspond to other compounds of Co.

### 3.2. Microstructure

Figure 3 shows the SEM image of the mixed powder of WC-based cemented carbide with a graphene content of 0.6 wt.%. As shown in Figure 3, graphene is a large particle sheet with a length of more than 2 μm. The graphene particles are evenly and tightly wrapped by the WC-based composite powder, and it can be observed that the WC particles in the composite powder are finer; that is, the addition of graphene helps to refine the WC grain size.

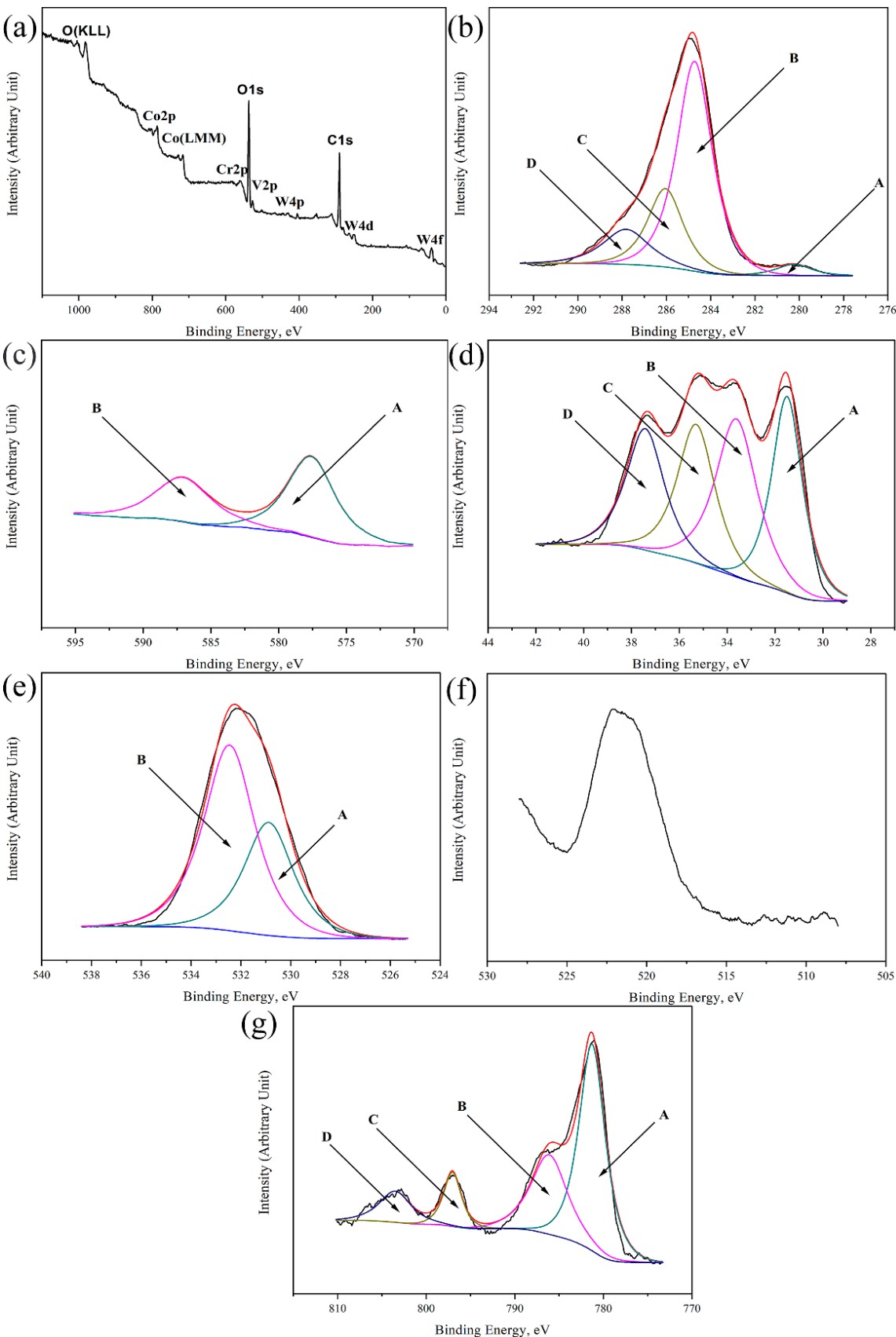

**Figure 2.** XPS spectra of WC-based cemented carbide with graphene content of 0.6 wt.% at 1350 °C, 8 min, and 50 MPa: (**a**) full spectrum; (**b**) C1s; (**c**) Cr2p; (**d**) W2p; (**e**) O1s; (**f**) V2p; (**g**) Co2p.

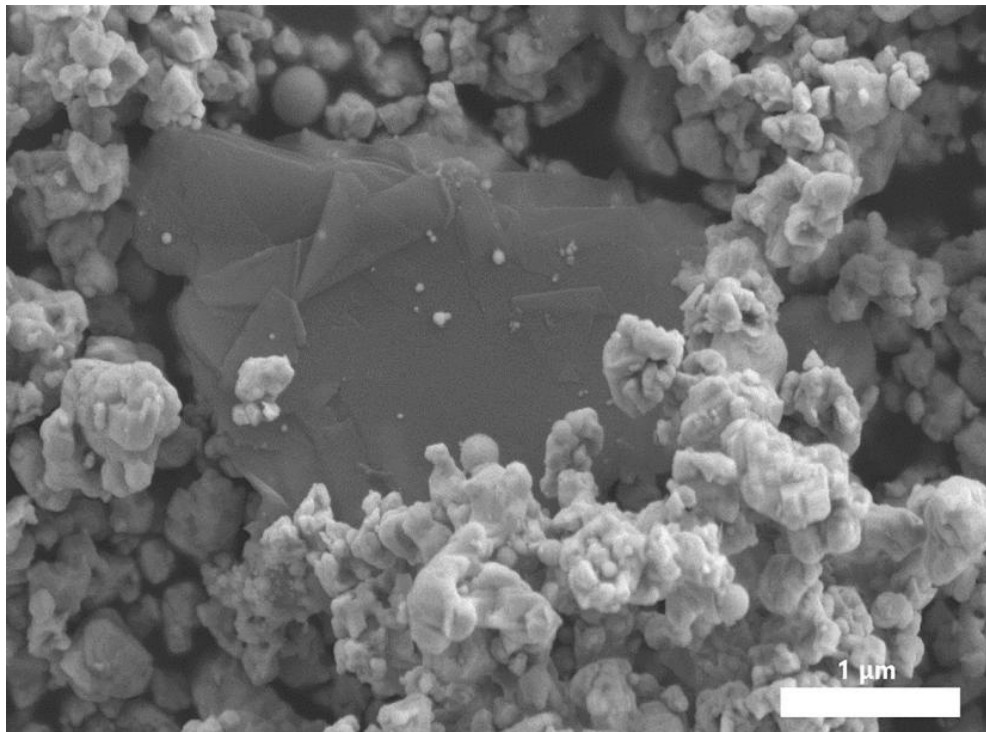

**Figure 3.** SEM image of mixed powder with 0.6 wt.% content of graphene.

Figure 4 shows the backscattering diagrams of cemented carbide with different contents of graphene at 1350 °C, 8 min, and 50 MPa and the distribution of each element. From this, it can be seen that there is a certain pattern of change in the microstructure of WC-based cemented carbide as the graphene content increases. In Figure 4a, the amount of graphene contained in the sample is 0.0 wt.%, the grain distribution is more uniform, the size is smaller, and the number of pores is small. The content of graphene in Figure 4b is 0.2 wt.%. After the addition of graphene, the WC-based cemented carbide produces a large number of pores, and the size of the pores is large. The generation of pores may be due to the low content of graphene, which is not uniformly dispersed in the WC-based cemented carbide matrix and has an effect on the penetration and diffusion of the Co bonding phase, which in turn leads to a large number of pores. As illustrated in Figure 4c, the content of graphene is 0.4 wt.%, the grain size of WC is still large, and the number and size of pores are slightly reduced. As shown in Figure 4d, when the graphene content increases to 0.6 wt.%, the size and number of pores in the sample decrease, and the grain size becomes finer, indicating that graphene was uniformly distributed on the WC matrix. In addition, the presence of graphene hinders the migration of WC grain boundaries, resulting in a smaller WC grain size [31]. Furthermore, in the composite materials prepared by spark plasma sintering, the addition of graphene can effectively hinder the growth of WC grains, improve the microstructure of the cemented carbides, and enhance the mechanical properties and magnetic properties of the matrix materials [32,33]. The graphene content continued to increase to 0.8 wt.%. As shown in Figure 4e, there is abnormal growth of some grains and graphene agglomeration, which may be due to the excessive graphene content and the aggregation of graphene leading to the aggregation of the Co phase, making the Co phase area increase and the refinement of the WC grains decrease, which has a certain effect on the microstructure, mechanical, and magnetic properties of the composite [6].

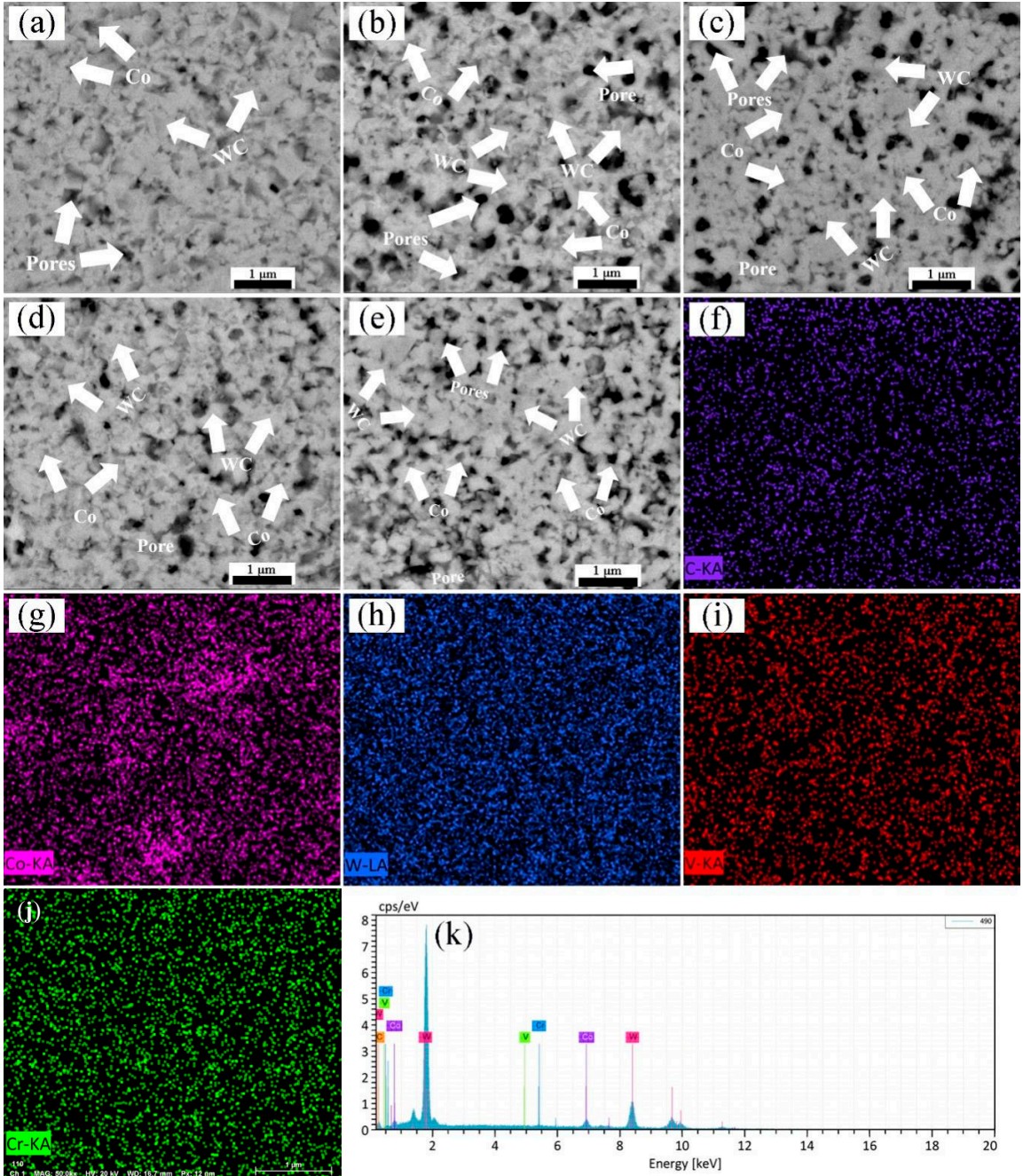

**Figure 4.** Microscopic morphology of WC-Co-Cr₃C₂-VC-graphene carbide with different graphene contents at 1350 °C, 8 min, and 50 MPa: (**a**) 0.0 wt.%; (**b**) 0.2 wt.%; (**c**) 0.4 wt.%; (**d**) 0.6 wt.%; (**e**) 0.8 wt.%; (**f**–**k**) mapping and EDS spectra of WC-based carbide with 0.6 wt.% graphene.

To observe the elements and distribution of the WC-based cemented carbides, a sample with a graphene content of 0.6 wt.% was performed by energy spectrum analysis (EDS), as shown in Figure 4f–k. It can be seen from the figure that the distribution of element C in the matrix is relatively uniform, and some of it is dissolved in the Co bonding phase, while the distribution of other elements, such as W, V, and Cr, in the sample is relatively uniform. As can be seen from Figure 4j, the main components of the selected area of the specimen are the elements C, W, and Co, as well as small amounts of V and Cr.

Figure 5 shows TEM and HRTEM images of cemented carbide with a graphene content of 0.6 wt.% sintered by SPS, allowing further observation of the microstructure and interface bonding of the sample. As can be seen in Figure 5a,c, the samples are mainly

composed of Co, WC, (W, V, Cr) $C_x$, and graphene, where (W, V, Cr) $C_x$ and graphene are mainly distributed between the WC/Co interface. It can be seen that the grain size of WC is 200–500 nm, and the phases are relatively uniformly distributed to form an organic combination since the incorporation of nano WC reduces the sintering temperature, leading to a prolonged solid–liquid transition time, which in turn facilitates the plastic flow of hard particles and particle rearrangement under high-temperature conditions [34]. The HRTEM image shows that the compound with a crystal plane spacing of 3.00 Å is WC and the compound with a crystal plane spacing of 1.92 Å is Co, the amorphous material corresponds to graphene, as shown in Figure 5b. The electron diffraction map in Figure 5a corresponds to (W, V, Cr) $C_x$, and the image in Figure 5c is a combination of electron diffraction clouds and fuzzy lattice forms for selected regions of graphene. This shows that during the SPS process, the surface of graphene reacts chemically with carbides or complexes to form new compounds. Due to the different solubility of the hard particles in the binding phase, the formation of (W, V, Cr) $C_x$ between the Co/WC interface helps to optimize the microstructure and overall properties of the sample. The incorporation of graphene played different degrees of bridging and toughening roles at different interfaces. Figure 5d is the EDS spectrum at the purple circle in Figure 5c, from which it can be analyzed that the added graphene is attached to the surface of the WC grain. Combined with the HRTEM diagram, it can be found that graphene can tightly bind Co and WC particles in the form of rotation and deformation and prevent the diffusion of element C from the WC to Co phase to react to form $Co_3C$. Graphene can also inhibit the mutual diffusion and growth of the Co phase so that the grain size is kept in a small range, thereby improving the fracture toughness of cemented carbide.

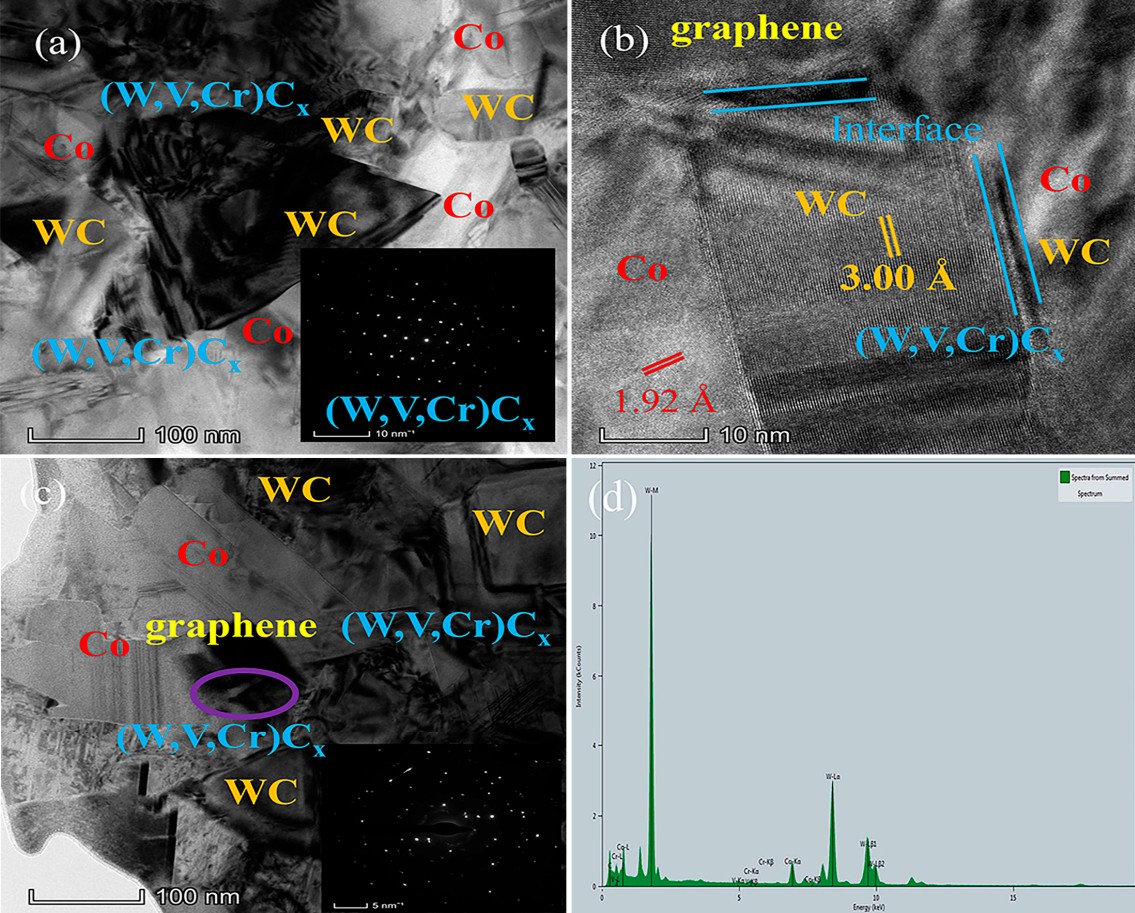

**Figure 5.** TEM and HRTEM images of cemented carbide(**a**–**d**) with graphene content of 0.6 wt.% at 1350 °C, 8 min, and 50 MPa.

Atomic force microscopy (AFM) observation was carried out on WC-based cemented carbide with a graphene content of 0.6 wt.% to investigate further the enhancement mechanism of graphene on WC-based cemented carbides sintered by SPS, as shown in Figure 6. The surface morphology and three-dimensional morphology of the sample can be seen in the figure. The structure of the specimen is relatively homogeneous, the grain size has been refined with the addition of graphene, and the number and size of pores have been greatly reduced.

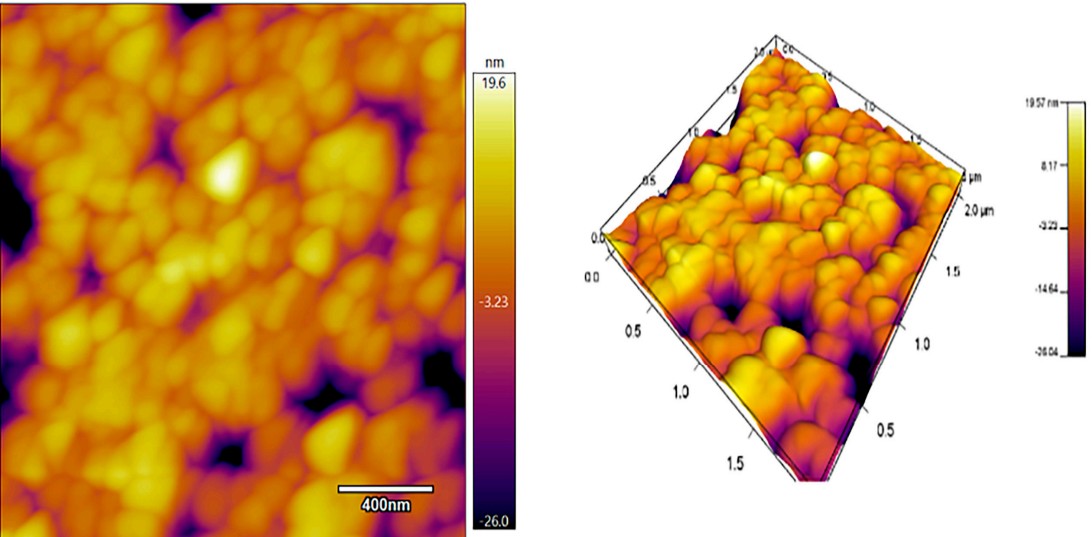

**Figure 6.** Atomic force micrograph of WC-based cemented carbide with graphene content of 0.6 wt.% at 1350 °C, 8 min, and 50 MPa.

*3.3. Mechanical Properties*

Figure 7 shows the effect of different levels of graphene on the mechanical properties of WC-based cemented carbide specimens. It can be seen from Figure 7 that under spark plasma sintering conditions, with the increase in graphene content, the density and hardness of cemented carbides show a trend of first increasing and then decreasing. When the graphene content is 0.0 wt.%, the hardness and relative density of cemented carbide are the highest, which are 2440 HV and 97.2%, respectively, and the fracture toughness of the specimen is 12.05 MPa·m$^{1/2}$ at this time. As shown in Figure 7, the relative density and hardness of the cemented carbide reach a maximum value of 94.3% and 2009 HV when the graphene content is 0.6 wt.%, and the fracture toughness of the specimen is 11.72 MPa·m$^{1/2}$ at this time. Studies have shown that the higher the grain density of WC and the finer the grain size, the more difficult it is to generate and move dislocations, thus improving the mechanical properties of composites [35]. In this study, this is also confirmed by the trend of hardness variation with the graphene content because the graphene particles are sheetlike, so it can effectively hinder the growth of the grain during the sintering process, thereby improving the mechanical properties of the composite. As the graphene content continues to increase, the hardness and density of the specimens show a decreasing trend, which may be due to the fact that the addition of graphene will affect the penetration and diffusion of the Co binding phase, resulting in the phenomenon of Co phase aggregation, thereby inhibiting the improvement of the mechanical properties of WC-based cemented carbide. It is also possible that because of the size of graphene particles at the nanoscale, with the increase in graphene addition, graphene itself is prone to agglomeration. The aggregated graphene not only reduces its adsorption capacity but also affects its excellent properties, resulting in the refinement of WC particles, which reduces the mechanical properties of WC-based cemented carbide [36].

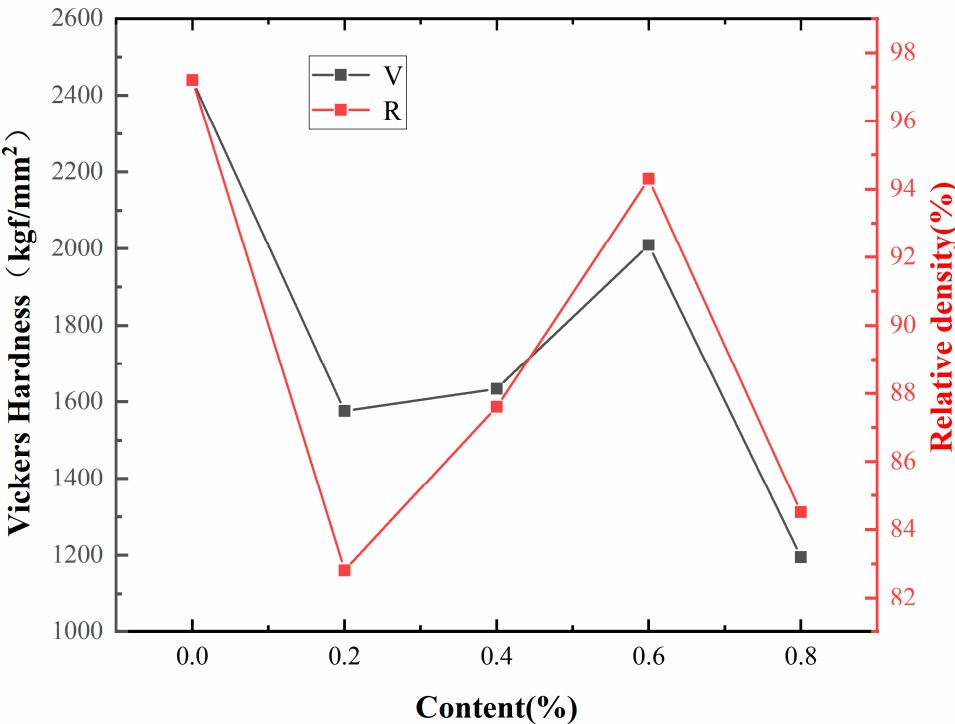

**Figure 7.** Relative density and hardness of cemented carbides with different content of graphene at 1350 °C, 8 min, and 50 MPa.

### 3.4. Hysteresis Loops and Magnetic Properties

Figure 8 shows typical hysteresis loops and magnetic properties of cemented carbides prepared by SPS under different contents. As shown in Figure 8, as the content of graphene increases, the saturation magnetization (Ms) and remanent magnetization (Mr) of cemented carbides show a downward trend, but the saturation magnetic induction (Hm) of alloys show a trend of first decreasing, then increasing, and then decreasing. The sample exhibits high saturation magnetization (8.95 emu/g) and residual magnetization (3.07 emu/g) at a graphene content of 0.0 wt.%. For the sample containing 0.6 wt.% of graphene, the saturation magnetization (Ms) is 7.78 emu/g, the remanent magnetization (Mr) is 1.77 emu/g, and the saturation magnetic induction (Hm) is 437.55 Oe. The Ms of the sample depends on the lattice structure and composition of the phases in the hard alloy. Due to the two-dimensional layered structure of graphene, its addition can, to some extent, increase the free path of hard phase dissolution in cobalt, reduce its solubility in cobalt, and thus improve its magnetic properties. However, because the presence of graphene may hinder the diffusion and penetration of Co, resulting in uneven dispersion, which affects the magnetic properties of the cemented carbide, the magnetic properties of the sample without the addition of graphene are better than that of the sample with the addition of graphene. For conventional magnetic materials, the smaller the crystal grains, the greater the coercive force the substance has; that is, the greater the coercive force, the smaller the grain size. Therefore, in the sample with graphene added, because the coercivity of the sample with a graphene content of 0.6 wt.% is the largest, its grain size is the smallest, and the finer the WC grain, the better the comprehensive performance of the sample.

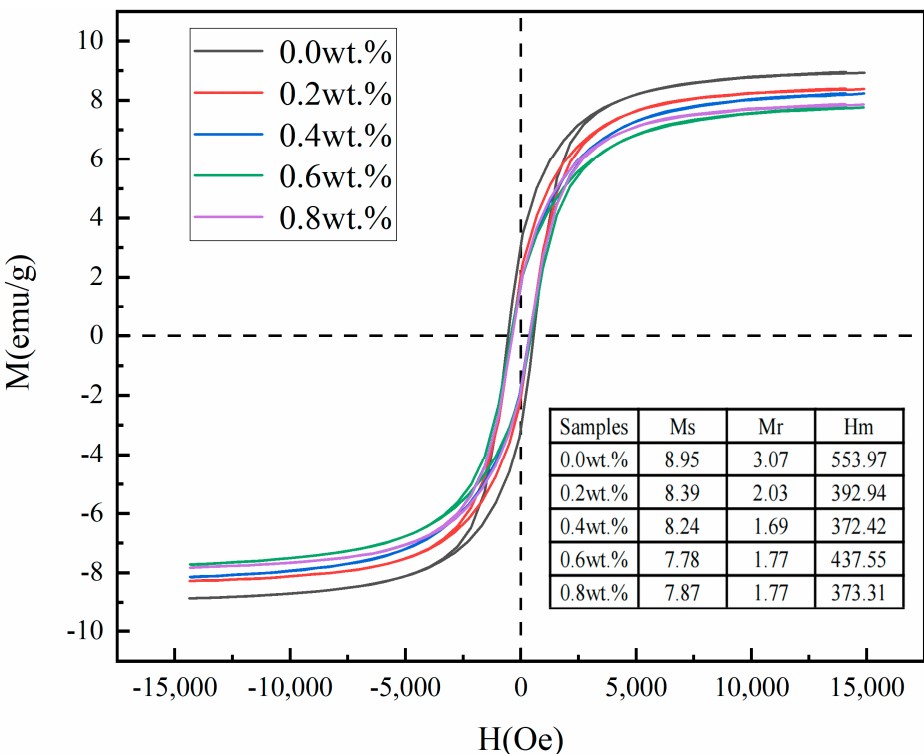

**Figure 8.** Hysteresis loops and magnetic properties of cemented carbide with different contents of graphene at 1350 °C, 8 min, and 50 MPa.

## 4. Conclusions

WC-based cemented carbide was prepared by spark plasma sintering at 1350 °C, 50 MPa, and 8 min. The effects of different graphene content on the microstructure, phase composition, mechanical properties, and magnetic properties of cemented carbide with different content of graphene were studied. When the graphene content is 0.0 wt.%, the relative density and hardness of the sample are 97.2% and 2440 HV, respectively, the fracture toughness is 11.72 MPa·m$^{1/2}$, and the coercivity is 553.97 Oe. With the increase in graphene content, the microstructure and mechanical properties of the samples show a trend of first increasing and then decreasing; the phase composition is basically unchanged, and the magnetic properties also have a certain tendency to change. When the graphene content is 0.6 wt.%, the relative density and hardness of the sample are 94.3% and 2009 HV, respectively, the fracture toughness is 11.72 MPa·m$^{1/2}$, and the coercivity is 437.55 Oe. The addition of graphene can inhibit the growth of WC grains in the process of SPS sintering, improve the mechanical properties of cemented carbide, and improve the microstructure of cemented carbide. However, excessive graphene will produce an agglomeration phenomenon, will also affect the penetration and diffusion of Co, and even cause Co phase aggregation, which weakens the inhibition effect on the growth of WC grains and then inhibits the improvement of the mechanical properties of WC-based cemented carbide.

**Author Contributions:** Conceptualization, Z.Z. and W.Q.; methodology, Z.Z. and W.Q.; software, X.Z. and X.L.; validation, Y.Q., H.Z. and S.Z.; formal analysis, Z.Z. and W.Q.; investigation, W.Q.; resources, Z.Z.; data curation, W.Q. and S.W.; writing—original draft preparation, W.Q.; writing—review and editing, Z.Z.; visualization, W.Q.; supervision, S.W. and S.Z.; project administration, Z.Z. and W.Q.; funding acquisition, Z.Z. All authors have read and agreed to the published version of the manuscript.

**Funding:** The research was sponsored by the Natural Science Foundation of China (52274362), the Key R & D projects of Henan Province (221111230800), the Innovative Funds Plan of Henan University of Technology (2021ZKCJ05), and the Science and Technology Collaborative Innovation Project of Zhengzhou (21ZZXTCX08), China.

**Data Availability Statement:** Data will be made available on request.

**Conflicts of Interest:** The authors declare no conflict of interest.

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
