# Peer review of "Effect of Graphene on the Microstructure and Mechanical Properties of WC-Based Cemented Carbide"

_crystals, doi:10.3390/cryst13101414_

Round 1

Reviewer 1 Report

In this paper WC-Co cemented carbides with varying addition of graphene and consolidated using SPS are studied. The influence of graphene addition on the resulting microstructure as well as properties such as density and hardness/fracture toughness is evaluated.

The introduction is well written and includes relevant papers. The relevance is pointed out in a concise manner. Please introduce the meaning of the abbreviations: GPL and GNS.

Section 2.2: the term "radiation range" is incorrect; 2 theta describes the angle. To  me "radiation range" implies that there is some range in the radiation wavelength.

Results, section 3.1: you state that no eta phase was detected, indicating that no reaction between graphene and WC-Co took place. I think that this is not really or only a matter of whether a reaction can take place kinetically - if you introduce carbon in the form of graphene, then it is unlikely that eta phase is formed.  Eta phase occurs to the left of the 'carbon window', i.e. if there is carbon deficiency. If you add carbon then the system is either in the two phase region WC-Co or at some point cross over to the 3-phase-region WC-Co+free carbon.

XRD results: there is an inset showing some slight shift of the WC peak. This is not discussed in the text, why is it included? Also the 2 theta values of the x-axis are very hard to read in the inset.

XPS results: you state that in Fig 2b the peak D corresponds to 'other carbides'. Which carbides?

The biggest flaw in the experimental plan is that no WC-Co reference without graphene addition is included.  This reference would give a lot of insight how the presence of graphene impacts the microstructure,  porosity, density, hardness... and would make it much easier to interpret the results. The influence of graphene addition on WC grain refinement, porosity and hardness can only be evaluated properly if you include a WC-Co-0% graphene sample. I suggest you strongly consider including such a sample. Without including a reference sample with no graphene addition at all, the conclusion ' graphene can [...] improve the mechanical properties of cemented carbide' is not supported by the presented data.

In Fig4 a/b/c it is clear that there is some porosity in the samples. It would be interesting to see if the same level of porosity occurs in a WC-Co sample without graphene addition. The SPS parameters should be optimized in such a way that 100 % dense samples are obtained. Porosity has a large impact on the properties of cemented carbides and conventional cemented carbides have a relative density of >99.5 %. Therefore it is quite common to calculate the relative density and I suggest that you include the relative density in addition to the measured density. I estimate that your theoretical density for your composition is e.g. 14.4 g/cm³ for the WC-Co-0.6% graphene sample. This would mean that the relative density is 94 %.

You state that graphene addition refines the WC grain size. The microstructure images in Fig 4 of the EDS mapping have low magnification, so it is not possible to estimate the grain size there. Do you have comparable AFM measurements to quantify the WC grain size and prove the refinement? Classic measurement of WC grain size with the linear intercept methode is probably too difficult due to the very small WC grain size. Another possibility is to measure coercive force, which gives an indirect measurement of WC grain size. Again, the WC grain size without any graphene addition would give great insight to how graphene addition influences grain growth.

Can you please elaborate what you mean with "penetration and diffusion of Co", this is unclear to me.

Please round the value of the measured hardness. Two decimal places are not appropiate considering the error of measurement.

Small remark regarding formatting: the font size of the authors is inconsistent.

Overall the paper is well written and easy to follow.
Line 47:  "introducing graphene as reinforced phase" --> this should be "reinforcing phase"

Author Response

Manuscript Number: crystals-2624207   

Effect of graphene on the microstructure and mechanical properties of WC-based cemented carbide

Thank you very much for your detailed review of our manuscript. Thank you for your insightful comments and suggestions. Based on them, we have revised the manuscript as follows. If there are still some problems in the manuscript please write to us. We will be very grateful to you for your help.

Best regards!

Yours sincerely,

Wanzhen Qi, Zhiwei Zhao, Yanju Qian, Shijie Zhang, Hongjuan Zheng, Xiaomiao Zhao, Xinpo Lu and Shun Wang

Name: Zhiwei Zhao

E-mail: [email protected] (Z.W. Zhao)

1.The introduction is well written and includes relevant papers. The relevance is pointed out in a concise manner. Please introduce the meaning of the abbreviations: GPL and GNS.                                                         

 Response: The GPL is a graphene platelet and the GNS is the graphene nanosheet. Thank you for pointing this out. We agree with this comment. Therefore, we have added the meaning of GPL and GNS respectively in line 46 of the second paragraph of the introduction on the first page and line 73 of the first section of Materials and Methods on the second page.       

 2. Section 2.2: the term "radiation range" is incorrect; 2 theta describes the angle. To me "radiation range" implies that there is some range in the radiation wavelength.                                                                     

Response: Thank you for pointing this out. We agree with this comment. Therefore, we have turned the sentence “An X-ray diffraction analyzer type D8AA25, introduced from Germany, was used for the physical phase analysis of WC matrix composites with Cu-Kα in the radiation range of 20°≤2θ≤90°.” into “An X-ray diffraction analyzer type D8AA25, introduced from Germany, was used for the physical phase analysis of WC matrix composites with Cu-Kα in the 2 theta angle range of 20°≤2θ≤90°.”, which is located at the second section of Materials and Methods in lines 83 to 85. 

3. Section 3.1: you state that no eta phase was detected, indicating that no reaction between graphene and WC-Co took place. I think that this is not really or only a matter of whether a reaction can take place kinetically - if you introduce carbon in the form of graphene, then it is unlikely that eta phase is formed.  Eta phase occurs to the left of the 'carbon window', i.e. if there is carbon deficiency. If you add carbon then the system is either in the two phase region WC-Co or at some point cross over to the 3-phase-region WC-Co+free carbon.

Response: The sentence “In agreement with Sun et al.'s studies [25-27], no harmful compounds such as η-phase were detected in all samples, indicating that there was no significant chemical reaction between graphene and the cemented carbide matrix.” has been revised as “In agreement with Sun et al.'s studies [27-29], no harmful compounds such as η-phase and free carbon were detected in all samples, indicating that the graphene and carbon content in the matrix is appropriate.”, which is located at the first section of Results and discussion in lines 121 to 123.

4. XRD results: there is an inset showing some slight shift of the WC peak. This is not discussed in the text, why is it included? Also the 2 theta values of the x-axis are very hard to read in the inset.

Response: With the increase of graphene content, the peak of WC has a slight shift. Thank you for pointing this out. We agree with this comment. We have discussed this in the text, “As can be seen from the inset of Fig. 1, with the increase of graphene content, the peak value of WC shifts first to the right and then to the left slightly. When the graphene content is increased from 0.0 wt.% to 0.4 wt.%, the peak value of WC shifts to the right slightly, according to Bragg's equation, the interplanar crystal spacing of WC grains decreases. When the graphene content is 0.6 wt.%, the peak value begin to shift to the left, indicating that after the addition of graphene, the crystal face spacing of the WC grains increases. This is because the solid solution strengthening effect occurs after graphene is added.”, which is located in the first section of Results and discussion in lines 108 to 116. We also added the sample data with a graphene content of 0.0 wt.% to the XRD pattern and highlighted the coordinates in the illustration for a closer look. 

5. XPS results: you state that in Fig 2b the peak D corresponds to 'other carbides'. Which carbides?

Response: Peak C (286.05 eV) and peak D (287.85 eV) may correspond to carbides of W, V, or Cr. We have discussed this in the text, which is located in the first section of “Results and discussion” in lines 137 to 138.

 6. The biggest flaw in the experimental plan is that no WC-Co reference without graphene addition is included.  This reference would give a lot of insight how the presence of graphene impacts the microstructure, porosity, density, hardness... and would make it much easier to interpret the results. The influence of graphene addition on WC grain refinement, porosity and hardness can only be evaluated properly if you include a WC-Co-0% graphene sample. I suggest you strongly consider including such a sample. Without including a reference sample with no graphene addition at all, the conclusion ' graphene can [...] improve the mechanical properties of cemented carbide' is not supported by the presented data.

Response: We have added the data of the sample with 0.0 wt.% graphene content to the XRD pattern, the micromorphology pattern, and the mechanical properties diagram for reference. We have discussed this in the text, which is located in the highlighted section of the adjacent paragraph of each graph.

 7. In Fig4 a/b/c it is clear that there is some porosity in the samples. It would be interesting to see if the same level of porosity occurs in a WC-Co sample without graphene addition. The SPS parameters should be optimized in such a way that 100 % dense samples are obtained. Porosity has a large impact on the properties of cemented carbides and conventional cemented carbides have a relative density of >99.5 %. Therefore it is quite common to calculate the relative density and I suggest that you include the relative density in addition to the measured density. I estimate that your theoretical density for your composition is e.g. 14.4 g/cm³ for the WC-Co-0.6% graphene sample. This would mean that the relative density is 94 %.

Response: Thank you for pointing this out. We agree with this comment. The relative density has been calculated. We have revised the manuscript accordingly. 

8. You state that graphene addition refines the WC grain size. The microstructure images in Fig 4 of the EDS mapping have low magnification, so it is not possible to estimate the grain size there. Do you have comparable AFM measurements to quantify the WC grain size and prove the refinement? Classic measurement of WC grain size with the linear intercept methode is probably too difficult due to the very small WC grain size. Another possibility is to measure coercive force, which gives an indirect measurement of WC grain size. Again, the WC grain size without any graphene addition would give great insight to how graphene addition influences grain growth.

Response: For the grain size of WC, we tested the hysteresis loop and magnetic properties of the samples. Thank you for pointing this out. We agree with this comment. We have discussed this in the text, which is located in the fourth section of Results and discussion in lines 266 to 287.  

9. Can you please elaborate what you mean with "penetration and diffusion of Co", this is unclear to me.

Response: During the sintering process, because graphene is dispersed in sheets between the Co/WC interface, it will hinder the diffusion and penetration of Co, resulting in the phenomenon of Co phase aggregation. Thank you for pointing this out. We agree with these comments. We have discussed this in the text, which is located in the fourth section of Results and discussion in lines 256 to 257.  

10. Please round the value of the measured hardness. Two decimal places are not appropiate considering the error of measurement.

Response: Thank you for pointing this out. We agree with this comment. The measured hardness value has been rounded off. We have revised the manuscript accordingly. 

11. Small remark regarding formatting: the font size of the authors is inconsistent.

Response: Thank you for pointing this out. We agree with this comment. We have revised the manuscript accordingly. 

12. Line 47:  "introducing graphene as reinforced phase" --> this should be "reinforcing phase"

Response: Thank you for pointing this out. We agree with this comment. The sentence “introducing graphene as reinforced phase” has been revised as “introducing graphene as reinforcing phase”, which is located in the first section of Introduction in line 52.

Reviewer 2 Report

In this manuscript, the authors present a detailed investigation of the microstructure and mechanical properties of graphene-containing WC-based cemented carbides prepared by spark plasma sintering (SPS).  The strategy of mixing graphene in WC-based cemented carbides and similar hard materials is useful and it is explored well in this work bringing a good understanding of the physics behind the role of graphene for the mechanical properties of the mentioned compounds. Thus, the present work is quite original and comes very timely. The characterization efforts (XRD, TEM, hardness measurements), and their concrete interpretation and realization are carefully discussed, explained, and presented in all the necessary detail. Consequently, the results are both original and realistically applicable to a wide range of similar aspects that may be explored in hard materials when graphene is introduced, thus inspiring future work in the field.

The figures, their captions and their corresponding discussion in the main text are easy to understand and they are logically organized too.

This work certainly represents a valuable contribution with possible wider impact to the field.

There are only minor concerns about textual details of this already excellent work that should be addressed before the manuscript becoming suitable for publication, i.e., it can be considered for publication after a minor revision:

1: In the title, the abbreviation SPS should not be included (SPS is not widely enough recognized abbreviation). In fact, the whole phrase “prepared by SPS” could be dropped from the title in the interest of a shorter and concise title.

2: Abstract should contain a brief but explicit reference to the characterization techniques employed.

3: The authors could provide a more detailed background when they mention “densification at lower temperatures” in connection to  References to just “higher” and “lower” temperatures should be avoided. Instead, explicit temperature ranges should be always provided.

4: In the introduction, the authors miss that in previous works dedicated theoretical methods such as (ab initio) molecular dynamics to understand similarly complex material phenomena related to graphene and in connection to thermal and mechanical properties, namely [Physical Chemistry Chemical Physics 25 (2023) 829-837; and Nanotechnology 33 (2022) 335706] have been widely used. Such works should be acknowledged.

Spell-check and stylistic revision of the paper are necessary. Some misspellings, etc., are noticeable throughout the text.

Author Response

Manuscript Number: crystals-2624207    

Effect of graphene on the microstructure and mechanical properties of WC-based cemented carbide

Thank you very much for your detailed review of our manuscript. Thank you for your insightful comments and suggestions. Based on them, we have revised the manuscript as follows. If there are still some problems in the manuscript please write to us. We will be very grateful to you for your help.

Best regards!

Yours sincerely,

Wanzhen Qi, Zhiwei Zhao, Yanju Qian, Shijie Zhang, Hongjuan Zheng, Xiaomiao Zhao, Xinpo Lu and Shun Wang

Name: Zhiwei Zhao

E-mail: [email protected] (Z.W. Zhao)

1.In the title, the abbreviation SPS should not be included (SPS is not widely enough recognized abbreviation). In fact, the whole phrase “prepared by SPS” could be dropped from the title in the interest of a shorter and concise title.

Response: Thank you for pointing this out. We agree with this comment. The whole phrase “prepared by SPS” has been dropped from the title to obtain a shorter and concise title. 

2. Abstract should contain a brief but explicit reference to the characterization techniques employed.

Response: Thank you for pointing this out. We agree with this comment. The sentence “The phase composition, microstructure, mechanical properties, and magnetic properties of cemented carbide were investigated by means of XRD, SEM, Vickers hardness and fracture toughness tests, and magnetic properties tests.” has been added to the Abstract, which is located in lines 9 to 12. 

 3. The authors could provide a more detailed background when they mention “densification at lower temperatures” in connection to  References to just “higher” and “lower” temperatures should be avoided. Instead, explicit temperature ranges should be always provided.

Response: Thank you for pointing this out. We agree with this comment. The sentence “Therefore, the raw materials of cemented carbide can be densified at lower temperatures (1200 ~1350℃) and shorter time (5~10 min)” has been added to the manuscript, which is located in the first section of Introduction in lines 60 to 62.

 4. In the introduction, the authors miss that in previous works dedicated theoretical methods such as (ab initio) molecular dynamics to understand similarly complex material phenomena related to graphene and in connection to thermal and mechanical properties, namely [Physical Chemistry Chemical Physics 25 (2023) 829-837; and Nanotechnology 33 (2022) 335706] have been widely used. Such works should be acknowledged.

Response: Thank you for pointing this out. We agree with this comment. The sentence “Moreover, from the perspective of dynamics, the epitaxial growth of heterogeneous structures of 2D graphene materials has the potential to make it a more excellent material with comprehensive properties such as mechanical properties and thermal stability [20,21].” has been added to the introduction, which is located in the first section of Introduction in lines 43 to 46.

Round 2

Reviewer 1 Report

Thank you for including the WC-Co-0% graphene data as suggested.

All my comments have been sufficiently addressed.

Throughout the manuscript now the unit of hardness is before the value. Please change it back to how it was before (e.g. 2000 HV and not HV 2000).